# Consumption of Sweet Beverages and Cancer Risk. A Systematic Review and Meta-Analysis of Observational Studies

**DOI:** 10.3390/nu13020516

**Published:** 2021-02-04

**Authors:** Fjorida Llaha, Mercedes Gil-Lespinard, Pelin Unal, Izar de Villasante, Jazmín Castañeda, Raul Zamora-Ros

**Affiliations:** Unit of Nutrition and Cancer, Cancer Epidemiology Research Programme, Catalan Institute of Oncology (ICO), Bellvitge Biomedical Research Institute (IDIBELL), 08908 Barcelona, Spain; fllaha@idibell.cat (F.L.); mgill@idibell.cat (M.G.-L.); pelin.uenal@dkfz-heidelberg.de (P.U.); idevillasante@idibell.cat (I.d.V.); acastaneda@idibell.cat (J.C.)

**Keywords:** systematic review, meta-analysis, cohort, case-control, sugar-sweetened beverages, artificial sweetened beverages, fruit juice, cancer

## Abstract

The consumption of sweet beverages, including sugar-sweetened beverages (SSB), artificial-sweetened beverages (ASB) and fruit juices (FJ), is associated with the risk of different cardiometabolic diseases. It may also be linked to the development of certain types of tumors. We carried out a systematic review and meta-analysis of observational studies aimed at examining the association between sweet beverage intake and cancer risk. Suitable articles published up to June 2020 were sourced through PubMed, Web of Science and SCOPUS databases. Overall, 64 studies were identified, of which 27 were selected for the meta-analysis. This was performed by analyzing the multivariable-adjusted OR, RR or HR of the highest sweet beverage intake categories compared to the lowest one. Random effects showed significant positive association between SSB intake and breast (RR: 1.14, 95% CI: 1.01–1.30) and prostate cancer risk (RR: 1.18, 95% CI: 1.10–1.27) and also between FJs and prostate cancer risk (RR: 1.03, 95% CI: 1.01–1.05). Although the statistically significant threshold was not reached, there tended to be positive associations for the following: SSBs and colorectal and pancreatic cancer risk; FJs and breast, colorectal and pancreatic cancer risk; and ASBs and pancreatic cancer risk. This study recommends limiting sweet beverage consumption. Furthermore, we propose to establish a homogeneous classification of beverages and investigate them separately, to better understand their role in carcinogenesis.

## 1. Introduction

The consumption of sweet beverages has increased in the last decades, with sugar-sweetened beverages (SSB) and artificially sweetened beverages (ASB) among the most widely consumed [1,2]. SSBs contain high levels of sugar that usually come from added sucrose or high fructose corn syrup (HFCS). Another type of sweet beverage is fruit juice (FJ), including fresh and commercial FJs and nectars. Despite their natural and healthy image, they contain high levels of sugar in the form of fructose. Although whole fruit also contains fructose, the fiber present limits the insulin response and increases satiety [3]. High sugar consumption may contribute to excessive energy intake, leading to long-term weight gain [4], higher risk of type 2 diabetes [5] and cardiovascular disease [6].

It has been demonstrated that obesity and type 2 diabetes are well-known risk factors for cancer [7,8,9]. Diets high in added sugar usually result in weight gain and an increase in adiposity-related metabolic parameters, insulin resistance, bioactivity of steroid hormones, oxidative stress and inflammation, which finally leads to cancer development and progression [9]. The International Agency for Research on Cancer (IARC) reported as strong evidence that excess body fat is a major risk factor for many cancers, including esophageal, pancreatic, colorectal, post-menopausal breast, endometrial, renal, ovarian, gallbladder, hepatic and gastric cardia, among others [10].

High sugar intake impairs glucose and insulin tolerance and augments insulin and insulin-like growth factor (IGF) levels. Insulin and IGF are major determinants of proliferation and apoptosis, and may therefore influence carcinogenesis [11]. Beverages high in sugar, including SSBs and FJs, have high glycemic indexes [12] which is also suggested to be linked to cancer [13]. Moreover, both caloric and noncaloric sweet palatable substances have been demonstrated to activate the dopaminergic reward system. This can trigger addictive-like behaviors, which might be responsible for increased body fat [14]. ASBs contain low or non-caloric sweeteners (e.g., aspartame) and have been marked as healthier alternatives to SSBs. However, some studies have suggested that ASBs are also deleterious as regards obesity [15] and type 2 diabetes risk [5]. Moreover, it has also been suggested that long-term consumption of aspartame, used in many ASBs, might be carcinogenic [16]. Aspartame in liquids can quickly break down into methanol, and the subsequent metabolized formaldehyde is a documented carcinogenic substance [17].

In light of all this evidence, the association between consumption of sweet beverages and cancer risk has been investigated and reviewed by different studies. A meta-analysis from 2014 studied the association between SSB/ASB consumption and overall and specific cancer but no links were found [18]. Likewise, a 2019 meta-analysis did not find any significant association between SSB/ASB intake and pancreatic cancer risk [19]. However, the two mentioned studies did not perform a separate analysis of SSBs and ASBs which might have elucidated their particular role on cancer. A pooled analysis from 2012 [20] suggested a modest positive association between SSB intake and the risk of pancreatic cancer. Another similar study from 2010 [21] showed no significant association with colon cancer risk. A qualitative review of longitudinal studies from 2018 [22] reported inconsistent results for SSB/FJ intake and cancer risk. A recent French publication [23] reported a positive association between FJs and overall cancer risk. Regarding ASB intake, their results for breast, colorectal and prostate cancer risk were nonsignificant. However, another study [24] showed an increased risk for leukemia in the total population as well as for non-Hodgkin lymphoma and multiple myeloma in men only.

Evidence suggests that the link between sweet beverages consumption and cancer onset is biologically plausible. However, each type of beverage may have different mechanisms of action and different roles in cancer onset. Therefore, our study aimed to investigate these associations, by conducting separate analyses for SSB, ASB and FJ intake and cancer incidence. We analyzed case-control and cohort studies and performed a meta-analysis when feasible. Through this study we intend to update and develop a better understanding of the association between the consumption of sweet beverages and cancer incidence, a disease that caused 9.6 million deaths in 2018, a figure projected to nearly double by 2040 [25].

## 2. Materials and Methods

### 2.1. Search Method for Identification of Studies

This study was conducted according to the Preferred Reporting Items for Systematic Reviews Meta-Analysis (PRISMA) guidelines. To identify the suitable articles, we searched in PubMed, Web of Science and SCOPUS databases up to 31 June 2020, using the following keywords: (((((“soft drinks”[All Fields] OR “sugary drinks”[All Fields]) OR “sugary beverages”[All Fields]) OR “fruit juice”[All Fields]) OR “sugar-sweetened beverages”[MeSH Terms]) OR “artificially sweetened beverages”[MeSH Terms]) AND ((((“neoplasms”[MeSH Terms] OR “neoplasm”[All Fields]) OR “cancer”[All Fields]) OR “cancers”[All Fields]) OR “tumor”[All Fields]). We also applied search filters by article type (excluding books, reviews, systematic reviews and meta-analyses) and by species (including only humans). Moreover, reference lists of included manuscripts and relevant reviews were examined for any possible unidentified study. The search process was limited to English and Spanish languages.

### 2.2. Eligibility Criteria and Data Extraction

Eligible cohort and case-control studies were selected if they met the following criteria: (1) included adult participants free of cancer (if prospective) or with no history of previous cancer (if case-control) at recruitment, except for nonmelanoma skin cancer; (2) overall or site-specific cancer incidence as an outcome; and (3) estimated and reported hazard ratio (HR), risk ratio (RR) or odds ratio (OR) with 95% confidence interval (CI) for the link between any type of sweet beverages and any type of cancer incidence. The exclusion criteria were: (1) participants with previous cancer history or currently undergoing cancer treatment; (2) cancer survival and cancer mortality as an outcome; and (3) duplicated studies. The following data were extracted: first author’s name, publication year, study name, country, age and sex of the participants, study sample size, number of cases and controls, follow-up duration, cancer site, type of exposure and amount of intake, dietary assessment methods, confounders’ adjustment and HR/RR/OR with 95% CI for the larger degree of adjustment. When time-varying results were reported, those related to baseline data were extracted.

Three review authors independently performed the literature search, study selection and data extraction (FL, MG-L, and PU). Disagreements were discussed between all authors until a consensus was reached.

### 2.3. Quality Assessment of Included Studies

Two independent review authors (FL and MG-L) examined the methodological quality of the individual studies using the Risk Of Bias In Non-randomized Studies—of Exposures (ROBINS-E) [26] tool for cohort studies and the Newcastle–Ottawa Scale (NOS) [27] adapted for case-control studies. The ROBINS-E tool evaluates the risk of bias by assessing different domains: confounding variables, selection of participants into the study, classification of exposures, departures from intended exposures, missing data, measurement of outcomes and selection of the reported result. Low, moderate or serious risk of bias was established in each study considering all domains. The NOS assesses the selection of groups (0–4 stars), adequacy of comparability between groups (adjustment for confounders) (0–2 stars) and ascertainment of the exposure of interest for case–control studies (0–3 stars). For selection domain, we considered studies with 0–1, 2–3 and 4 stars as serious bias risk, moderate bias risk and high-quality risk, respectively. For comparability between groups, we considered those with 0, 1 and 2 as serious, moderate, and low bias risk, respectively. And finally, for ascertainment of exposure, we considered those 0, 1–2 and 3 as serious risk, moderate risk and low bias risk, respectively. In both tools, when data were not enough for judgment, the domain was classified as ‘no information’.

### 2.4. Data Synthesis and Statistical Analysis

The first obstacle that we had to overcome was the lack of a unique definition for beverages and a variety of other terms. In this text, the following group terms are used to generalize these products: SSB for sugar-sweetened beverages (regular soft drinks/sodas, and non-diet soft drinks/sodas), ASB for artificially sweetened beverages (low and noncaloric soft drinks/sodas, and diet soft drinks/sodas) and FJ for fruit juices. In addition, two other terms are used: SB for sweetened beverages that includes both SSBs and ASBs; SFJ for high-sugar (added or natural) beverages that includes both SSBs and FJs. The quantity of each beverage was provided mostly as categories of frequency of consumption, either in amount (mL or g/day) or serving sizes (cans for SSBs and ASBs, glasses for FJs). To unify the data, we converted the categories to mL/day, based on the study-specific serving size for each beverage. When the serving size was not reported, we referred the national data of each study. Thus, we considered one can equal to 330 mL and one glass equal to 200 mL for European countries [28], one can equal to 360 mL and one glass equal to 240 mL for the United States [29], and one can equal to 375 mL for Australia [30]. One US study [31] expressed consumption as grams of sugar, and we weighed up an average of 10.5 g of sugar per 100 mL of SSB and an average of 9.6 g of sugar per 100 mL of FJ. This was calculated based on the sugar content of different commercially available products of popular brands [32].

Prior to the analysis, the selected studies were classified by outcome (cancer incidence by site) and exposure (SB, SSB, ASB, FJ and SFJ). Data were summarized in a narrative manner and a meta-analysis was performed only if at least three studies reported data for the same exposure and outcome. In the meta-analysis, results for the total number of participants were considered. Separate analyses were considered (e.g., European-American and African-American women) when the article did not report indices for total population. In the same manner, if studies reported data for specific beverages (e.g., caffeinated and non-caffeinated SSBs), results for the total beverage group (e.g., total SSBs) were weighted up. Despite having extracted data on fruit and vegetables juices together, for the meta-analysis we considered the studies that indicated FJs as the predominant beverage consumed. The meta-analysis was performed by pooling the multivariable-adjusted RR/HR/OR of the highest category of the exposure versus the lowest one, and random effects models were assumed. If statistical outliers were identified, secondary analyses were performed (without outliers) to remove possible sources of heterogeneity. An outlier was considered when its 95% CI lied outside the 95% CI of the pooled effect. To further explain heterogeneity, we performed subgroup and sensitive analyses, dividing studies according to design (cohort/case-control), country (US/non-US, mostly European), level of overall risk of bias (serious/low-moderate) and beverage intake category (high vs. non-consumer/high vs. low). We used Cochran’s Q, I^2^ and Tau^2^ statistics to measure between-study heterogeneity. The statistical analysis was performed with the Metafor package [33] of the R software, version 4.0.1. *P* values < 0.05 were considered statistically significant.

## 3. Results

### 3.1. Literature Search and Study Characteristics

The study selection process according to PRISMA guidelines is reported in Figure 1. In total, 869 potential publications were identified from the databases (PubMed, Web of Science and SCOPUS) and other sources. After removing duplicates, 596 articles were selected, from which 435 were excluded based on titles and 26 on abstracts. Of 135 eligible articles, 71 were excluded due to the following reasons: 59 did not report risk index for sweet beverages and cancer incidence, 3 full-texts were not available, 7 considered other outcomes, 1 case-control study included controls with cancer at recruitment and 1 publication was not in English or Spanish. Finally, 64 studies were included in the systematic review, 27 cohort [23,24,28,31,34,35,36,37,38,39,40,41,42,43,44,45,46,47,48,49,50,51,52,53,54,55,56] and 37 case-control studies [57,58,59,60,61,62,63,64,65,66,67,68,69,70,71,72,73,74,75,76,77,78,79,80,81,82,83,84,85,86,87,88,89,90,91,92,93]. Of these, 27 studies were meta-analyzed.

Of the included studies, 29 were performed in the United States (US), 17 in Europe, 6 in Asia, 5 in Canada, 3 in Australia, 2 in Latin-America, 1 in Egypt and 1 was multinational (Italy, Spain, Poland, Northern Ireland, India, Cuba, Canada, Australia and Sudan). They usually included both male and female participants. Ages ranged from 18 to 97 years. The 27 cohort studies were published between 2003 and 2020 and enrolled 4,458,056 participants in total, of which 30,646 developed cancer. Mean duration of the follow-up in cohort studies varied from 2 to 20 years. The 37 case-control studies were published between 1985 and 2019. In total, they enrolled 20,827 cancer cases and 34,315 controls. Most of the controls were selected from the general population.

Sweet beverage consumption in both cohort and case-control studies was expressed as categorical or continuous variables. Exposure assessment was collected using food frequency questionnaires (FFQ), 24-h dietary recalls (24-H DR), dietary questionnaires (DQ), interviews, or surveys. Among all the studies, 37 types of cancer were considered as an outcome and 4 cohorts reported data for overall cancer risk, including different types of cancer [23,50,52,54]. In most of the studies, the outcome was confirmed by a medical diagnosis. Overall characteristics of the included studies are summarized in Table 1. Results of the meta-analysis for the random-effect model are summarized in Table 2 and for the subgroup analysis in Appendix A.

### 3.2. Sweet Beverages and Risk of Breast Cancer

Nine publications reported data on breast cancer, four case-control [57,58,69,80] and five cohort studies [23,51,52,53,55]. In the meta-analysis with six publications, including four cohort studies [23,52,53,55] and two case-controls [57,80], a significant positive association between high SSB consumption and breast cancer risk was observed (RR: 1.14, 95% CI: 1.0–1.3) (Table 2). No associations were found for FJ intake (Table 2). Marzbani et al. [58] reported a positive association with SBs (OR: 2.8, 95% CI: 1.9–4.3), but no associations were found for ASBs. Subgroup analyses for SSB consumption did not explain further heterogeneity (Appendix A).

#### 3.2.1. Sweet Beverages and Risk of Pre-Menopausal Breast Cancer

Three cohort publications [23,53,55] and one case-control (taken as two as indices were separated by ethnicity) [57] were included in the analysis of SSB intake and pre-menopausal breast cancer. Their pooled analysis showed a borderline statistically non-significant positive association (RR: 1.37, 95% CI: 0.99–1.88) (Appendix A), which reached the significance in the subgroup analysis including only cohort studies (RR: 1.60, 95% CI: 1.08–2.37) (Appendix A). A cohort study from 2019 [23] also reported data for ASB, FJ and SFJ intake and only indicated a positive association for SFJs (HR: 1.28, 95% CI: 1.09–1.83).

#### 3.2.2. Sweet Beverages and Risk of Post-Menopausal Breast Cancer

A meta-analysis of four cohort studies [23,53,54,55] and one case-control (taken as two as indices were separated by ethnicity) [57] of SSBs showed non-significant results (Table 2). We performed subgroup analyses based on study design, country, and beverage intake categories. No statistically significant results were found from the heterogeneity test between groups (Appendix A). Chazelas et al. [23] investigated the relationship with SFJ consumption and observed a positive association (HR: 1.44, 95% CI: 1.05–1.99). No significant results were reported for ASBs.

### 3.3. Sweet Beverages and Risk of Intestinal and Colorectal Cancer

Eight publications reported data on colorectal cancer, four case control [88,89,90,91] and four cohort studies [23,52,54,56]. A borderline positive association was observed with SSB intake using the random-effect model (RR: 1.18, 95% CI: 0.99−1.41) (Appendix A). No significant results were found either for SBs or for FJs (RR: 2.02, 95% CI: 0.45−9.01 (SB); RR: 0.79, 95% CI: 0.16−3.87 (FJ) (Appendix A). After the exclusion of one outlier, results for the random-effect model remained non-significant. No associations were found for colorectal cancer risk and ASBs. With regard to rectal cancer, no associations were observed with ASBs, SSBs or fruit and vegetables juices [92]. A case-control study on small intestine cancer [65] indicated a significant positive association with SSB consumption (OR: 3.6, 95% CI: 1.3−9.8).

### 3.4. Sweet Beverages and Risk of Esophageal Cancer

Three publications, one cohort [34] and two case-control studies [59,93] reported data on different types of esophageal cancers, including esophagus-gastric junction, esophageal adenocarcinoma and squamous cell carcinoma. No significant associations were shown between SB, SSB and ASB consumption and esophageal cancers risk.

### 3.5. Sweet Beverages and Risk of Gastric Cancer

One case-control [59] and two cohort studies [34,54] reported data on different types of gastric cancer (overall, cardia and non-cardia) and SBs, ASBs or SSBs showing no significant associations.

### 3.6. Sweet Beverages and Risk of Pancreatic Cancer

Eleven publications, six cohort [41,42,43,44,45] and five case-control studies [76,77,78,79,81] reported data on pancreatic cancer. No significant results were observed for SBs, SSBs or ASBs (Table 2). Although high heterogeneity was observed for SBs (I^2^ = 58.6, *p* = 0.02) and ASBs (I^2^ = 43.6, *p* = 0.13) (Table 2), after performing subgroup analyses results slightly improved but remained non-significant (Appendix A). No association was observed between FJ intake and pancreatic cancer risk.

### 3.7. Sweet Beverages and Risk of Genitourinary Cancer

#### 3.7.1. Bladder

Six case-control studies [60,61,62,63,64,65] reported data on bladder cancer. No association between SB consumption and bladder cancer risk was observed in the random-effect meta-analysis including five case-control studies [61,62,63,64,65] (Appendix A). We observed a high heterogeneity in the meta-analysis (I^2^ = 83.4%, *p* = 0.0001). Although heterogeneity was reduced after excluding outliers and doing subgroup analyses, the associations were positive but non-significant (Appendix A). A US study suggested a statistically significant relation between SB intake and bladder cancer risk [65]. Two case-control studies [60,65] also considered SSBs and ASBs separately. In a Chinese case-control study [62], SSB intake was suggested as a risk factor for bladder cancer, although no association was found for FJs. Similarly, in a Serbian study [63], no significant association was observed between FJs and bladder cancer risk.

#### 3.7.2. Prostate

Eight publications, six cohorts [23,31,35,36,52,54] and two case-controls [66,67] showed data on prostate cancer. No significant associations were reported for SBs from quantitative analysis. However, positive relations were observed in the random-effect model for SSBs (RR: 1.18, 95% CI: 1.10−1.27) and FJs (RR: 1.03, 95% CI: 1.01−1.05). The results remained the same in a subgroup analysis with 3 non-US (France, Spain, Australia) studies (RR: 1.13, 95% CI: 1.03−1.24) (Appendix A). Two cohorts [23,54] reported data on ASB intake and only one [23] found an increased prostate cancer risk of 33% (HR: 1.33, 95% CI: 1.01−1.75).

#### 3.7.3. Renal and Urothelial Cell Cancer

Four publications, two case control [68,70] and two cohort studies [37,54] provided data on renal cell cancer. For our meta-analysis, we selected three publications, two case-control [68,70] and one control study [37] on SBs, but the random-effect meta-analysis showed non-significant results (Table 2). Despite observing a high heterogeneity (I^2^ = 65.4%, *p*-value = 0.058), no outliers were found, and the number of studies was too low to perform subgroup analyses (n = 3). One case control study [70] reported a positive association with the intake of ASBs (OR: 2.7, 95% CI: 1.1−6.5) but not the other two [37,54]. No significant results were reported for SSBs or FJs, despite one case-control [68] finding a positive association with the consumption of fruit and vegetable juices taken together (OR: 1.53, 95% CI: 1.18−1.99). The EPIC cohort study [38] reported data on urothelial cell cancer and its association with SBs and FJs. A significant positive association was found only with FJ intake (HR: 1.32, 95% CI: 1.05−1.66).

### 3.8. Sweet Beverages and Risk of Gynecological Cancers

Two case-control studies [71,72] investigated the relationship between FJ intake and cervical cancer risk. Only one of them [72] found an inverse association (RR: 0.3, 95% CI: 0.2−0.6). Two cohort studies [39,54] reported data on different types of beverages (SSBs, ASBs, FJs and SFJs) and endometrial cancer risk. Only one of them [39] found significant positive associations with both SSBs (HR: 1.78, 95% CI: 1.32−2.40) and SFJs (HR: 1.48, 95% CI: 1.09−2.00). Finally, three case-control studies [69,70,71] reported data on epithelial ovarian cancer risk. Only one of them [71] found positive associations for caffeinated (OR: 1.51, 95% CI: 1.03−2.22) and non-caffeinated SBs (OR: 2.60, 95% CI: 1.25−5.36). No significant associations were reported for ovarian cancer risk [50].

### 3.9. Sweet Beverages and Risk of Hepatobiliary Cancers

Two cohort studies [28,49] reported data on different types of sweet beverages and various types of hepatobiliary cancers. The EPIC cohort [28] found no significant results regarding the consumption of either SBs or FJs and biliary tract cancer risk. However, a positive association was observed between both SBs (HR: 1.89, 95% CI: 1.11−3.02) and FJs (RR: 1.03, 95% CI: 1.01−1.06) and hepatocellular carcinoma risk. The Swedish Mammography Cohort and the Cohort of Swedish Men [49] found significant positive associations with both gallbladder (HR: 2.24, 95% CI: 1.02−4.89) and extrahepatic biliary tract cancer risks (HR: 1.79, 95% CI: 1.02−3.13). No significant results were reported for intrahepatic biliary tract cancer risk.

### 3.10. Sweet Beverages and Risk of Hematologic Cancers

One cohort study [24] reported data on leukemia and multiple myeloma and its association with SSB and ASB intake. Significant associations were found between the consumption of ASBs and leukemia risk (RR: 1.42, 95% CI: 1.00−2.02). No associations were observed in two cohorts [24,40] as regards SSBs or ASBs and non-Hodgkin lymphoma risk.

### 3.11. Sweet Beverages and Risk of Upper Aerodigestive Cancers

Four studies [34,82,83,84] reported data on upper aerodigestive cancers. One US-based cohort [34] showed no significant association between SB intake and pharyngeal, laryngeal and oral cavity cancer risks. A case-control study from Montenegro [82] suggested an inverse relation between SBs and larynx cancer risk. The consumption of FJs was inversely associated with oral cavity cancer risk in one case-control study [83] though not in another [84].

### 3.12. Sweet Beverages and Risk of Other Cancers

Single studies reported data on different types of cancer and their link with sweet beverages. No significant associations were reported for cutaneous melanoma [85], glioma [47] or thyroid cancer risk [48] and any type of sweetened beverages. One case-control study [86] reported an inverse association between natural juices (fruit and vegetables) and lung cancer risk (OR: 0.3, 95% CI: 0.3−0.4).

### 3.13. Sweet Beverages and Risk of Overall Cancer

An Australian cohort [50] investigated the association between SSBs and ASBs and the risk of non-obesity-related cancers; they reported a positive association only with ASBs (HR: 1.23, 95% CI: 1.02−1.48). Two cohorts [23,52] assessed the relationships between the intake of several types of sweet beverages and obesity-related cancer risk. Only one of them [23] showed positive associations with SSBs (HR: 1.06; 95% CI: 1.02−1.21), FJs (HR: 1.14, 95% CI: 1.01−1.29) and SFJs (HR: 1.30, 95% CI: 1.17−1.52). No association was found for ASBs and obesity-related cancer risk.

### 3.14. Quality of Included Studies

According to the ROBINS-E tool (Figure 2a, Appendix A), 13 of 27 cohort studies presented a moderate overall risk of bias. This is due to some bias being detected mostly in the classification of the exposure domain, deviation from the intended intervention and missing data. Missing data bias was not evaluated for 5 cohorts [36,39,43,51,52], as the publications did not report enough information. All studies fulfilled the criteria of low risk of bias for selection of participants’ domain. In addition, 3 [36,37,54] of 27 studies did not adjust the statistical analysis for all potential confounders. Therefore, they were classified at moderate risk of bias. Only one study [50] was classified as moderate risk of bias for outcome measurement, and another [56] for the selection of reported outcomes.

According to the NOS (Figure 2b, Appendix A) most of the case-control studies (29 of 37) presented a moderate overall risk of bias; 7 publications presented a serious risk, whereas 1 indicated a low risk. The risk of bias due to the selection of the groups was classified as moderate for 35 studies, high for 2 [58,82] and low for another 2 [59,66]. Most of the case-control studies adjusted their results for relevant and additional confounders and were classified as moderate or low risk of bias for comparability between groups. In addition, 5 were considered as serious risk for this domain, because 4 of them did not adjust for all important confounders [60,63,66,92] and 1 [79] reported results from an unadjusted analysis. Moreover, 5 studies [81,86,88,89] did not report this information and were classified as ‘no information’ category. The risk of bias due to ascertainment of the exposure was considered moderate in all case-control studies.

## 4. Discussion

### 4.1. Association between Consumption of Sweet Beverages and Cancer Risk

The aim of this study was to assess the relationships between different groups of sweet beverages and site-specific or overall cancer risk. We conducted a meta-analysis when at least three studies reported data for the same exposure (sweet beverage type) and outcome (cancer site). We found several statistically significant and borderline positive associations between the consumption of SBs, especially SSBs, and in some cases ASBs or FJs, and several cancer risks.

Regarding breast cancer, the meta-analysis showed a positive association using random effects, with a 14% higher risk for SSBs, but non-statistically significant results for pre- and post-menopausal breast cancer. However, after performing subgroup analyses by study type, cohort studies showed significant positive results for pre-menopausal breast cancer and SSBs. Chazelas et al. [23] reported a positive linear trend between SSB intake and breast/pre-menopausal breast cancer risk when SSB consumption increased by 100 mL/day. In line with our results, current evidence supports the World Cancer Research Fund/American Institute for Cancer Research (WCRF/AICR) recommendations of reducing or avoiding SSB intake for breast cancer prevention [94]. One US case-control study [57] conducted a separate analysis for African-American and European-American women. This showed a positive link between SSB intake and post-menopausal breast cancer risk for European-American women only. Likewise, two other cohorts that included mostly Caucasian women [53,54] showed similar results. This evidence suggests that ethnic differences may play a role. However, we could not explore this association as no other studies included women of African descent. In fact, evidence on the role of nutritional factors in breast cancer for this population is limited and inconclusive [95]. Our meta-analysis did not find significant associations between FJs and breast cancer risk. With regards to the SFJ group, comparing highest versus lowest consumption, Chazelas et al. [23] reported positive relations for SFJs and total, pre- and post-menopausal breast cancer risk. Conversely, Makarem et al. [52] showed no significant associations. A publication from the US [69] found no positive associations for SBs and breast cancer risk; however, a recent case-control study [58] found positive associations.

For colorectal cancer risk, our meta-analysis found no positive results using random effects for SB, SSB or FJ intake. Despite having performed secondary analyses excluding outliers and having explained between-studies heterogeneity, results for the random-effect model remained non-significant. This is in a way consistent with results from a previous meta-analysis, which found no association between SSBs and colon cancer risk using a random-effects model [21]. On the other hand, a cohort study from 2014 found a positive association for an increase in 330 mL/day of SSBs [91]. Likewise, an Australian study that compared extreme categories of SSB intake (≥200 mL/day versus <6.7 mL/day) showed positive results [54]. We included only one study assessing rectal cancer incidence [92]. Here, a separate analysis for women and men was performed. The majority of the results were not significant, and the only positive association was found for juice (fruit and vegetables) consumption in female participants.

In regard to esophageal cancers, publications included in this review were also part of a meta-analysis from 2014 [18]. This meta-analysis reported no association between SBs and esophageal adenocarcinoma and squamous cell carcinoma risk. After extracting separated data for SSB and ASB intake, we found similar results. Despite these observations, positive associations were found in a pooled analysis of US-based case-control studies. This study assessed the association between sugar dietary intake and Barret’s esophagus incidence, a precursor for esophageal adenocarcinoma tumor [96]. Even though data from the included studies reported non-significant results for stomach cancer incidence, a Japanese cohort study observed that carbonated drinks and juices appeared to be related to an elevated risk of death from stomach cancer [97].

With respect to pancreatic cancer, we performed a meta-analysis for SBs, SSBs and ASBs. These associations, especially for SBs, tended to be positive but did not reach statistically significant levels using random effect models. These results go along with a recent meta-analysis from 2019 [19] which also showed no association between SB intake and pancreatic cancer risk. Besides that, a pooled analysis from 2012 [20] reported a 56% higher risk of pancreatic cancer for males consuming ≥375 mL/day of SSBs compared to non-consumers. Likewise, a Swedish cohort [41] found a 93% higher risk of pancreatic cancer incidence among those who consumed ≥500 mL/day of SSBs compared to non-consumers. However, we performed a subgroup analysis taking into account beverage intake category (high vs. non- consumer), but no significant associations were observed (Appendix A). In addition, only one study reported separate results for carbonated and noncarbonated SBs, but no significant results were shown [76].

For bladder cancer risk, 3 out of the 6 included case-control studies [62,63,65] showed positive associations for highest versus lowest amounts of SB intake. However, the meta-analysis of these studies together with 2 other case-control studies [61,64] showed no significant associations. Despite performing a second analysis excluding one study that presented some serious bias, the results remained non-significant (Appendix A). Hence, our meta-analysis of observational studies reported that SBs appeared to be unrelated to bladder cancer risk. It is not clear how SSBs, ASBs or FJs act in isolation as the evidence is limited.

With reference to prostate cancer, our meta-analysis demonstrated an 18% higher risk for SSBs comparing the highest with the lowest intake. Similarly, we found a small positive association for FJs (a 3% higher risk). No associations were found for SBs, which may suggest that the role of ASBs might not be relevant. However, one study [23] reported a positive association between ASB intake and prostate cancer risk.

Renal cell cancer appeared to be unrelated to SB consumption according to the meta-analysis results. We observed a high between-study heterogeneity (I^2^= 65.4%). However, not enough studies (n = 3) were included to perform subgroup analyses. Even so, Maclure and Willet [70] reported a significant positive association between highest versus lowest SB intake and renal cell cancer risk (RR: 2.6, 95% CI: 1.4−4.8). More studies analyzing this association are required for further clarification.

The association between SSB consumption and both endometrial and ovarian cancer risk tended to be positive but did not reach statistically significant levels. One study stratified results by types of endometrial cancer (I and II) [39]. They reported positive associations between highest versus lowest SSB and SFJ consumption and endometrial type I cancer in post-menopausal women, but not in type II. These might be because subtypes may have different risk factors, even though evidence on this etiologic heterogeneity is quite limited [98]. Data from two studies [71,72] suggested that FJ intake might be a protective factor for cervical cancer. FJ consumption is often considered part of a healthy diet and lifestyle [99]. However, none of the mentioned studies [71,72] adjusted for such confounders. Thus, it is not clear if the protective effect was due to FJ intake or other factors. For epithelial ovarian cancer, one US study [75] stratified the results by caffeinated and non-caffeinated colas. Both results were positive statistically significant, but non-caffeinated colas showed a stronger association. Although this might suggest a protective effect of caffeine, a recent meta-analysis of prospective studies found no link between caffeine intake and ovarian cancer risk [100].

In respect of hepatobiliary cancers, data from the included studies showed a positive association with SB consumption, especially for gallbladder cancer, where the risk was doubled [49]. This might be explained by the detrimental association between sucrose/glycemic load and the increased risk of symptomatic gallstone disease [101], which is strongly correlated with gallbladder cancer [102]. Stepien et al. [28] showed slightly positive dose–response associations between SBs, ASBs or FJs and HCC incidence.

As regards hematologic cancers, no associations were found either for sugary or for artificially sweetened beverages, except for leukemia risk, for which one study [24] reported significant positive associations with ASBs. However, a recent review of clinical trials and observational studies observed no association between artificial sweeteners intake and both leukemia and non-Hodgkin lymphoma incidence [103].

The evidence is more limited regarding cancer of the oral cavity, pharynx, larynx, lung, thyroid, glioma and cutaneous myeloma. The available data mainly showed nonsignificant results for SB and FJ intake. Only one study from Montenegro indicated an inverse association between SB intake and laryngeal cancer risk [82]. However, the results from this study should be treated with caution as they presented some methodological inadequacies and its overall risk of bias was classified as ‘serious risk’ (Appendix A). One case-control study from 1997 observed a strong positive association between small intestine cancer risk and SSBs (OR: 3.6, 95% CI: 1.3−9.8), although further high quality evidence is needed [87]. One study that reported incidence of overall non-obesity-related cancers showed no association for SSBs but a positive association for ASBs [50]. Moreover, the largest of 3 studies [23,52,54] on overall obesity-related cancer risk showed positive associations with SFJs, SSBs and FJs, but not with ASB consumption [23]. Similarly, a meta-analysis of clinical trials and observational studies showed no association between artificial sweetener intake, body weight and different types of cancers [103]. Our findings are in accordance and we agree with the previous study [103] upon the uncertainty of the evidence that links artificial sweeteners with different types of cancer.

### 4.2. Limitations of the Current Data

To the best of our knowledge, this is the first systematic review to evaluate the isolated association between different groups of sweet beverages and cancer risk. Several limitations should be considered while interpreting our findings. Some studies included in this systematic review were difficult to compare due to their design (cohort and case-control studies), methodology, classification and categories of beverages intake. Therefore, it was a challenge to perform such comparisons. According to the ROBIN-E tool, cohort studies were at low-moderate risk of bias. As per the NOS, the case-control studies were at moderate risk of bias and 6 studies [58,60,66,79,82,86] out of 37 presented serious methodology inadequacies. The number of publications included in most meta-analysis was relatively low (between 3 and 6). On this basis, the pooled effect size was calculated based on risk ratios of cohort and case-control studies together. Not having enough studies was a major limitation to perform subgroup analyses when high between-study heterogeneity was observed. Moreover, the small amount of studies may have been a potential source of unexplained heterogeneity [104]. We did not have enough data to perform subgroup analyses based on different population characteristics (e.g., sex, lifestyle factors or history of cancer). However, we did perform subgroup analyses based on geographical area.

The majority of the included participants were from the US or European countries. Hence, extrapolating our findings to other geographical areas may not be appropriate. We attempted to classify beverages into specific groups. However, some studies did not report precise information on this topic, which might have given rise to misclassifications. Similarly, we attempted to convert original exposure information into amounts of intake (mL/day) based on national data. Nevertheless, this was not possible in all studies which prohibited performance of a dose–response meta-analysis. Another limitation may be the measurement error in collecting dietary data since self-reported questionnaires were used. Moreover, in the longitudinal studies we were limited to the baseline estimation of beverages consumption, and there is a possibility that their consumption changed over time. It is suggested that the link between SSBs or FJs and cancer risk is possible due to their high glycemic indexes [13] and to obesity-inducing pathways [4]. However, these variables were not adequately integrated as confounders in all the studies. Indeed, glycemic index was only considered in one cohort [53]. Despite BMI being a common indicator of obesity and most studies considering it as confounder, only 4 of them [35,40,52,54] adjusted for other obesity indicators such as waist circumference. Most of the studies assessed the association between consumption of SSB and common cancers such as breast, colorectal, prostate and pancreatic cancer. Data were more limited for FJs or ASBs and other types of cancers, especially non-obesity-related ones. FJ consumption may coexist with healthy habits, such as healthy diet or exercise [99]. Therefore, it would have been even better if some studies had adjusted their analysis for such variables. In fact, only 3 publications [52,53,54] used diet quality as a confounder.

## 5. Conclusions

The current meta-analysis of cohort and case-control studies indicated a statistically significant positive association between higher consumption of SSB and breast and prostate cancer incidence. As regard pre-menopausal breast cancer, results from cohort studies alone showed a significant association. Likewise, it showed a statistically significant positive link between high intakes of FJs and prostate cancer risk. Although the associations between other sweet beverages and other cancer types were also positive, they did not reach statistically significant levels. The small number of studies and cancer cases might have been a reason why we did not find statistically significant results for several cancer types. Study location (US/non-US, mostly European) did not appear to influence the results. Current evidence indicates that higher incidence of some cancers is related to a high consumption of SSBs. However, the evidence is limited to make recommendations regarding ASBs and FJs. This subject requires further investigation.

We encourage future research in this field to perform more separated analysis on SSB, ASB and FJ consumption. We believe it would be prudent to establish a homogeneous classification of beverages in order to better understand their role in carcinogenesis. We also recommend considering other obesity-related factors besides body mass index, such as waist circumference, glycemic index and quality of diet as confounders. We could not study the different roles of non-carbonated soft drinks (sport, fruit and tea-based drinks), sometimes used as healthier alternatives to carbonated drinks [105]. Therefore, it would be advisable for future studies to further explore this research area.

This systematic review supports the WCRF/AICR recommendations to limit sugary drinks consumption for cancer prevention [106] and to raise consumers’ awareness of their low nutritional quality and high sugar content. We recommend replacement of sweet beverages with plain safe drinking water and infusions without added sugars as the main liquid source for body hydration. Even though some guidelines maintain that moderate consumption of FJs may be part of a healthy diet [107], FJs contain little or no dietary fiber and are positively associated with tooth decay in children [108]. Professional societies have recently recommended limiting children’s FJ consumption as means of addressing the obesity epidemic [3]. Whole fruits and plain safe drinking water should also be affirmed as a healthier alternative to sweet beverages in adults. This would aim to promote the appropriate consumption of essential nutrients, to reduce intake of excessive sugars/calories and to therefore lower cardiometabolic disease and cancer incidences [109,110]. The increase in cancer [25], obesity [111] and type 2 diabetes [112] requires policy action. We recommend policymakers worldwide to consider (or continue with) taxation and marketing restriction for sweet beverages, especially SSBs.

## Figures and Tables

**Figure 1 nutrients-13-00516-f001:**
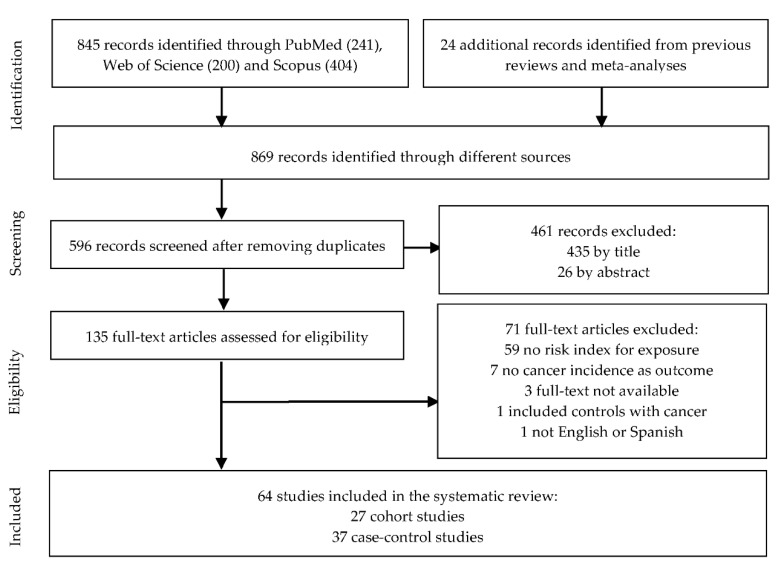
Prisma diagram.

**Figure 2 nutrients-13-00516-f002:**
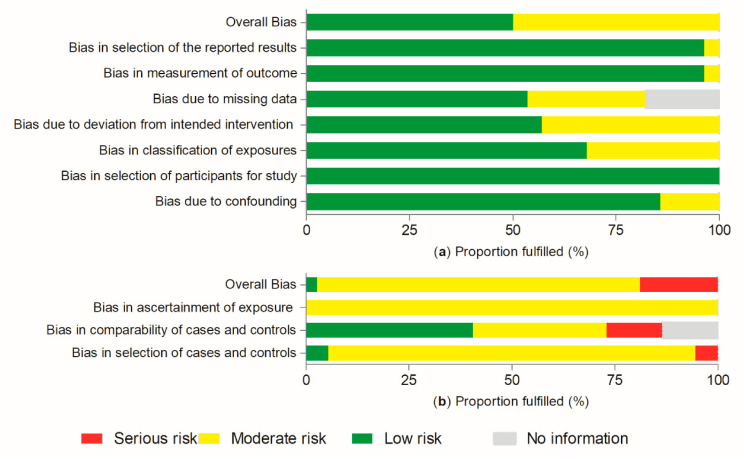
Risk of bias in the included studies. Legend: (**a**) risk of bias in cohort studies according to the Risk of Bias in Non-randomized Studies−of Exposures (ROBINS-E) tool and (**b**) risk of bias in case-control studies according to the Newcastle–Ottawa Scale (NOS).

**Table 1 nutrients-13-00516-t001:** Overall characteristic of the included studies.

	Breast Cancer (Breast, Pre- and Post-Menopausal)										
Source	Country, Study Name	Cancer Type	Study Design	Population Follow-Up (Years)	Cases	Age (Mean/SD or Range)	Sex (%)	Dietary Assessment Method	Type and Amount of Beverages Intake ^+^	HR/RR/OR (95% CI)	Adjustments
Chandran et al., 2006 [57]	US, WCHS	Breast	PB case-control	3148	1558	20–75	F (100)	125-item FFQ	SSB: ≥152 vs. <152 mL/day	OR: 0.97 (0.74–1.27) (AA) OR:1.31 (0.91–1.89) (EA) OR: 1.17 (0.79–1.74) (AA) OR: 0.95 (0.58–1.56) (EA) OR: 0.76 (0.51–1.12) (AA) OR: 2.05 (1.13–3.7) (EA)	Age, ethnicity, country, education, age at menarche, menopause and first birth, MS, parity, BF status, history of benign breast disease, family history of BC, HRT, OC use, BMI, and study site.
		Pre-M			797				SSB: ≥152 vs. <152 mL/day		
		Post-M			761				SSB: ≥152 vs. <152 mL/day		
Chazelas et al., 2019 [23]	France, NNS	Breast	Cohort	101,257 5.1 (median)	693	42.2/14.4	F (78)	24H-DR	SFJ: >123 vs. <38.1 mL/day (cut-off) SFJ: increase by 100 mL/day SSB: >57.1 vs. <13.6 mL/day (cut-off) SSB: increase by 100 mL/day ASB: >11.6 vs. <4.6 mL/day (cut-off) ASB: increase by 10 mL/day FJ: >81.9 vs. <17.0 mL/day (cut-off) FJ: increase by 100 mL/day SFJ: >123 vs. <38.1 mL/day (cut-off) SFJ: increase by 100 mL/day SSB: >57.1 vs. <13.6 mL/day (cut-off) SSB: increase by 100 mL/day ASB: >11.6 vs. <4.6 mL/day (cut-off) ASB: increase by 10 mL/day FJ: >81.9 vs. <17.0 mL/day (cut-off) FJ: increase by 100 mL/day SFJ: >123 vs. <38.1 mL/day (cut-off) SFJ: increase by 100 mL/day SSB: >57.1 vs. <13.6 mL/day (cut-off) SSB: increase by 100 mL/day ASB: >11.6 vs. <4.6 mL/day (cut-off) ASB: increase by 10 mL/day FJ: >81.9 vs. <17.0 mL/day (cut-off) FJ: increase by 100 mL/day	HR: 1.37 (1.08–1.73) HR: 1.22 (1.07–1.39) HR: 1.10 (0.87–1.39) HR: 1.23 (1.03–1.48) HR: 1.33 (0.98–1.75) HR: 0.97 (0.86–1.09) HR: 1.13 (0.91–1.39) HR: 1.15 (0.97–1.35) HR: 1.28 (1.09–1.83) HR: 1.26 (1.04–1.51) HR: 1.68 (1.45–1.74) HR: 1.34 (1.15–1.70) HR: 1.23 (0.52–2.53) HR: 0.95 (0.81–1.13) HR: 0.98 (0.67–1.43) HR: 1.10 (0.85–1.41) HR: 1.44 (1.05–1.99) HR: 1.19 (0.98–1.44) HR: 0.99 (0.72–1.39) HR: 1.08 (0.79–1.47) HR: 1.10 (0.55–2.12) HR: 1.01 (0.86–1.18) HR: 1.24 (0.95–1.61) HR: 1.19 (0.96–1.48)	Smoking, education, PA, BMI, and height.
		Pre-M			283						
		Post-M			410						
Hirvonen et al., 2006 [51]	France, SUVIMAX	Breast	Cohort	4396 6.6	95	35–60	F (100)	24H-DR	FJ: >150 mL/day vs. none	RR: 1.29 (0.80–2.09)	Age, smoking, number of children, OC use, family history of BC, and MS.
Makarem et al., 2018 [52]	US	Breast	Cohort	3184 4	128	54.3	F (53)	FFQ	SFJ: >324 vs. <135 mL/day (cut-off) SSB: >51.4 mL/day vs. none FJ: >180 vs. <38.6 mL/day (cut-off)	HR: 1.00 (0.65–1.57) HR: 1.04 (0.64–1.71) HR: 1.03 (0.67–1.62)	Age, smoking, BMI, EI, alcohol, PA, education, MS, nº of live births, WC, DM and CVD, antioxidant use, energy from fat, and diet soda intake.
Marzbani et al., 2019 [58]	Iran	Breast	HB case-control	620	212	40.2	F (100)	11-item healthcare form	SB ^7^: favorable intake vs. ≤1 time/month	OR: 2.8 (1.9–4.3)	Age, education, and BMI
McLaughlin et al., 1992 [69]	US	Breast	PB case-control	3234	1617	56.7	F (100)	SQ-interview	SB ^2^: ever vs. never	OR: 1.08 (0.92–1.26)	Age, alcohol, country, race, MS, age at first live birth, diagnosis of benign cancers, and family history of BC.
Potischman et al., 2002 [80]	US	Breast	PB case-control	2019	568	20–44	F (100)	100-item FFQ	SSB: ≥320 mL/day vs. none	OR: 1.09 (0.8–1.5)	Age at diagnosis, study site, race, education, alcohol consumption, years of OC use, smoking, BMI, and EI.
Romanos-Nanclares et al., 2019 [53]	Spain	Breast	Cohort	10,713 2	100	33.0 (median)	F (100)	FFQ	SSB: >47.1 vs. <11 mL/day	HR: 1.36 (0.74–2.50)	Age, height, family history of BC, smoking, PA, BMI, age at menarche and menopause, MS, HRT, number of pregnancies >6 month and before 30 years old, months of BF, alcohol, education, DM, GI, EI, U-P food and coffee consumption, and Med-diet adherence.
		Pre-M			57				SSB: ≥11 mL/day vs. none	HR: 1.16 (0.66–2.07)	
		Post-M			43				SSB: >47.1 vs. <11 mL/day	HR: 2.12 (1.01–4.41)	
Hodge et al., 2018 [54]	Australia, MCCS	Post-M	Cohort	35,593 19	946	54.6	F (100)	121-item FFQ	SSB: ≥200 vs. <6.7 mL/day ASB: ≥200 vs. <6.7 mL/day	HR: 1.11 (0.85–1.45) HR: 0.95 (0.73–1.25)	Socioeconomic indexes, country of birth, alcohol intake, smoking, PA, Med-diet score, and sex. ASB also for SSB consumption and WC.
Nomura et al., 2016 [55]	US, BWHS	Breast Pre-M Post-M	Cohort	49,103 13.8	1827 678 826	21–69	F (100)	FFQ	SSB: ≥250 mL/day vs. none SSB: ≥250 mL/day vs. none SSB: ≥250 mL/day vs. none	HR: 0.71 (0.50–1.02) HR: 1.72 (0.91–3.23) HR: 1.11 (0.77–1.61)	Age, geographic region of residence, EI, smoking, family history of BC, education, MS, OC use, parity, HRT, BMI, alcohol, PA, and sedentary time.
	**Colorectal and Rectal Cancer**										
**Source**	**Country, Study Name**	**Cancer Type**	**Study Design**	**Population Follow-Up (Years)**	**Cases**	**Age (Mean/SD or Range)**	**Sex (%)**	**Dietary Assessment Method**	**Type and Amount of Beverages Intake ^+^**	**HR/RR/OR** **(95% CI)**	**Adjustments**
Bener et al., 2010 [88]	Qatar	Colorectal	HB case-control	428	146	53.4	M (58)	DQ	SB: ≥330 vs. ≤47.1 mL/day	OR: 1.62 (1.19–2.17)	Not reported
Chazelas et al., 2019 [23]	France	Colorectal	Cohort	101,257 5.1 (median)	166	42.2 (14.4)	F (78)	24H-DR	SFJ: >123 vs. <38.1 mL/day (F); >141.7 vs. <46.1 mL/day (M) (cut-off) increase by 100 mL/day SSB: >57.1 vs. <13.6 mL/day (F); >65.5 vs. < 14.0 mL/day (M) (cut-off) increase by 100 mL/day ASB: >11.6 vs. <4.6 mL/day (F); >7.9 vs. < 2.7 mL/day (M) (cut-off) increase by 10 mL/day FJ: >81.9 vs. <17.0 mL/day (F); >97.8 vs. <19.9 mL/day (M) (cut-off) increase by 100 mL/day	HR: 1.07 (0.63–1.80)	Smoking, education, PA, BMI, and height.
										HR: 1.10 (0.84–1.46)	
										HR: 1.01 (0.59–1.71)	
										HR: 1.11 (0.72–1.71)	
										HR: 0.80 (0.44–1.46)	
										HR: 1.02 (0.94–1.10)	
										HR: 1.19 (0.78–1.82)	
										HR: 1.05 (0.75–1.46)	
Hodge et al., 2018 [54]	Australia, MCCS	Colorectal	Cohort	35,593 19	1055	54.6	M/F	121-item FFQ	SSB: ≥200 vs. <6.7 mL/day ASB: ≥200 vs. <6.7 mL/day	HR: 1.28 (1.04–1.57) HR: 0.79 (0.60–1.06)	Socioeconomic indexes, country, alcohol, smoking, PA, Med-diet score, and sex. ASB also for SSB consumption and WC.
Makarem et al., 2018 [52]	US	Colorectal	Cohort	3184 4	68	54.3	F (53)	FFQ	SFJ: >362.6 vs. <154.3 mL/day (cut-off) SSB: >180 vs. <25.7 mL/day (cut-off) FJ: >180 vs. < 48.9 mL/day (cut-off)	HR: 1.39 (0.68–2.82) HR: 0.96 (0.51–1.82) HR: 1.66 (0.88–3.12)	Age, smoking, BMI, EI, alcohol, PA, education, MS, nº of live births, WC, DM and CVD, antioxidant use, energy from fat, and diet soda intake.
Mahfouz et al., 2014 [89]	Egypt	Colorectal	HB case-control	450 1	150	<20–>60	F (52)	DQ	SB: daily vs. not daily FJ: daily vs. not daily	OR: 4.6 (1.9–11.01) OR: 0.18 (0.09–0.36)	Not reported
Pacheco et al., 2019 [56]	US	Colorectal	Cohort	99,798 20.1 (median)	1318	52.0 (13.5)	F (100)	FFQ	SSB: ≥60 mL/day vs. never/rare	HR: 1.14 (0.86–1.53)	Age, BMI, EI, smoking, alcohol, family history of CR polyps, multivitamin use, and HT.
Tayyem et al., 2018 [90]	Jordan	Colorectal	HB case-control	501 2	220	52	F (51)	Q-DQ	SB: daily vs. rarely OJ: daily vs. rarely	OR: 1.39 (0.73–2.63) OR: 1.07 (0.45–2.55)	Age, sex, work status, income, PA, marital status, EI, education, other diseases, and history of CR cancer.
Theodoratou et al., 2014 [91]	Scotland	Colorectal	PB case-control	4838 7.0	2062	64.3	M/F	FFQ	SSB: increase by 330 mL/day FJ: increase by 200 mL/day	OR: 1.12 (1.05–1.19) OR: 1.19 (1.11–1.27)	Age, sex, BMI, PA, family history of CR cancer, EI, NSAIDs, eggs, FJ, SSB, white fish, coffee, and magnesium intake.
Murtaugh et al., 2004 [92]	US	Rectal	PB case-control	2157 4	952	30–79	M (57)	Interview	SSB: yes vs. no (M) SSB: yes vs. no (F) ASB: yes vs. no (M) ASB: yes vs. no (F) J: >449 vs. ≤58.3 mL/day (M); J: >596.6 vs. ≤44.6 mL/day (F)	OR: 1.00 (0.80–1.26) OR: 0.96 (0.73–1.27) OR: 1.28 (0.98–1.68) OR: 0.90 (0.67–1.22) OR: 0.92 (0.63–1.34) OR: 1.56 (1.00–2.41)	Age, PA, EI, and dietary fiber and calcium intake.
	**Esophageal Cancers (Esophagus-Gastric Junction, Esophageal Adenocarcinoma, Squamous Cell Carcinoma)**										
**Source**	**Country, Study Name**	**Cancer Type**	**Study Design**	**Population Follow-Up (Years)**	**Cases**	**Age (Mean/SD or Range)**	**Sex (%)**	**Dietary Assessment Method**	**Type and Amount of Beverages Intake ^+^**	**HR/RR/OR** **(95% CI)**	**Adjustments**
Ibiebele et al., 2008 [93]	Australia	AEGJ	PB case-control	2341 4	325	18–79	M (71)	FF	SB ^7^: ≥375 mL/day vs. none SSB ^7^: yes vs. no ASB ^7^: yes vs. no SB ^7^: ≥375 mL/day vs. none SSB ^7^: yes vs. no ASB ^7^: yes vs. no SB ^7^: ≥375 mL/day vs. none SSB ^7^: yes vs. no ASB ^7^: yes vs. no	OR: 1.07 (0.67–1.73) OR: 0.63 (0.43–0.92) OR: 0.77 (0.46–1.29) OR: 0.94 (0.53–1.66) OR: 1.20 (0.79–1.81) OR: 0.71 (0.37–1.37) OR: 0.40 (0.20–0.78) OR: 0.70 (0.47–1.03) OR: 0.46 (0.25–0.85)	Age, sex, BMI, EI, alcohol, smoking, education, heartburn, and acid reflux symptoms.
		EAC			294						
		SCC			238						
Mayne et al., 2006 [59]	US	EAC	PB case-control	1782	228	65 Q1, 59.3 Q4	M (78 Q1, 82 Q4)	Proxy and self-interviewed	SSB ^7^: ≥355 vs. 10.7 mL/day	OR: 0.47 (0.29–0.76)	Age, sex, center, race, proxy interview status, BMI, EI, alcohol and meat intake, cigarettes/day, education, income, and frequency of reflux symptoms.
		SCC			206				SSB ^7^: ≥355 vs. 10.7 mL/day	OR: 0.85 (0.48–1.52)	
Ren et al., 2010 [34]	US, NIH-AARP-DHS	EAC	Cohort	481,563 2	305	50–71	M (59)	124-item FFQ	SB: ≥355 vs. ≤355 mL/day	HR: 1.11 (0.66–1.85)	Age, sex, smoking, alcohol, EI, BMI, education, ethnicity, PA, and daily intake of fruit, vegetables, red meat, and white meat.
		SCC			123				SB: ≥355 vs. ≤355 mL/day	HR: 0.85 (0.46–1.56)	
**Stomach Cancers (Gastric Cardia, Gastric Noncardia)**											
**Source**	**Country, Study Name**	**Cancer Type**	**Study Design**	**Population Follow-Up (Years)**	**Cases**	**Age (Mean/SD or Range)**	**Sex (%)**	**Dietary Assessment Method**	**Type and Amount of Beverages Intake ^+^**	**HR/RR/OR** **(95% CI)**	**Adjustments**
Hodge et al., 2018 [54]	Australia, MCCS	Gastric cardia	Cohort	35,593 19	165	54.6	M/F	121-item FFQ	SSB: ≥200 vs. <6.7 mL/day ASB: ≥200 vs. 6.7 mL/day	HR: 1.17 (0.73–1.89) HR: 1.03 (0.53–1.98)	Socioeconomic indexes, country, alcohol, smoking, PA, Med-diet score, and sex. ASB also for SSB consumption and WC.
Mayne et al., 2006 [59]	US	Gastric cardia Gastric noncardia	PB case-control	1782	255	65 Q1, 59.3 Q4	M (78 Q1, 82 Q4)	Proxy and self-interviewed	SSB ^7^: ≥355 vs. <10.7 mL/day	OR: 0.74 (0.46–1.16)	Age, sex, center, race, proxy interview status, BMI, EI, alcohol and meat intake cigarettes/day, education, incomes, and frequency of reflux symptoms.
					352				SSB ^7^: ≥355 vs. <10.7 mL/day	OR: 0.65 (0.43–0.98)	
Ren et al., 2010 [34]	US, NIH-AARP-DHS	Gastric cardia Gastric noncardia	Cohort	481,563 2	231	50–71	M (59)	124-item FFQ	SB: ≤355 vs. ≥355 mL/day	HR: 0.89 (0.55–1.45)	Age, sex, smoking, alcohol, EI, BMI, education, ethnicity, PA and daily intake of fruit, vegetables, and white meat.
					224				SB: ≥355 vs. ≤355 mL/day	HR: 0.75 (0.45–1.24)	
**Pancreatic Cancer**											
**Source**	**Country, Study Name**	**Cancer Type**	**Study Design**	**Population Follow-Up (Years)**	**Cases**	**Age (Mean/SD or Range)**	**Sex (%)**	**Dietary Assessment Method**	**Type and Amount of Beverages Intake ^+^**	**HR/RR/OR** **(95% CI)**	**Adjustments**
Bao et al., 2008 [42]	US, NIH-AARP-DHS	Pancreatic	Cohort	487,922 7.2	1258	50–71	F (41)	124-item FFQ	SB: 816.9 mL/day (median) vs. none SSB: 512.8 mL/day (median) vs. none ASB: 816.9 mL/day (median) vs. none	RR: 1.07 (0.86–1.33) RR: 1.01 (0.77–1.31) RR: 1.11 (0.86–1.44)	Age, sex, race, education, BMI, alcohol, smoking, PA, EI, and foliate intake. SSB and ASB were mutually adjusted.
Chan et al., 2009 [76]	US, SFB	Pancreatic	PB case-control	2233	532	21–85	M (53)	131-item FFQ	SB: ≥355 mL/day vs. none SB ^7^: ≥355 mL/day vs. none SSB ^7^: ≥355 mL/day vs. none ASB ^7^: ≥355 mL/day vs. none SSB ^4^: ≥355 mL/day vs. none	OR: 1.0 (0.7–1.3) OR: 1.1 (0.8–1.5) OR: 0.9 (0.6–1.3) OR: 1.5 (1.1–2.1) OR: 1.0 (0.6–1.8)	Age, sex, EI, BMI, race, education, smoking, history of DM, PA, red and white meat, fruit and vegetables, eggs, dairy, whole and refine grained, and sweets. SSB and ASB were mutually adjusted.
Gallus et al., 2011 [77]	Italy	Pancreatic	HB case-control	978 7	326	63 (median)	M (53)	FFQ	SB ^7^: ≥150 vs. <150 mL/day	OR: 1.02 (0.72–1.44)	Age, sex, study center, education, BMI, smoking, alcohol, EI, family history of pancreatic cancer, and DM.
Gold et al., 1985 [78]	US	Pancreatic	HB, PB case-control	676	274	66.1	F (53)	Interview	ASB: ever vs. never	OR: 0.66 (0.38–1.2)	Religion, occupation, smoking, and alcohol.
Larsson et al., 2006 [41]	Sweden, SMC, COSM	Pancreatic	Cohort	77,797 7.2	131	60.8	F (45)	FFQ	SB: ≥500 mL/day vs. none	HR: 1.93 (1.18–3.14)	Age, sex, education, smoking, BMI, and EI.
Lyon et al., 1992 [79]	US	Pancreatic	PB case-control	512	149	40–79	M/F	DQ	SB (caff): ever vs. never	OR: 1.31 (0.89–1.94)	Unadjusted.
Mack et al., 1986 [81]	US	Pancreatic	PB case-control	980	490	18–65	M (58)	Proxy and direct Interview	SB ^7^: ≥1650 vs. <1320 mL/day	RR: 2.6 (0.9–7.4)	Not reported
Mueller et al., 2010 [43]	China and Singapore, SCHS	Pancreatic	Cohort	60,524 14	140	56.5	F (56)	FFQ	SB: ≥67.7 mL/day vs. none J ^5^: ≥67.7 mL/day vs. none	HR: 1.87 (1.10–3.15) HR: 1.31 (0.74–2.30)	Age, sex, smoking, BMI, alcohol, EI, PA, DM, education, added sugar, and candy. SB and J were mutually adjusted.
Nothlings et al., 2007 [44]	US	Pancreatic	Cohort	162,150 8	434	59.8	F (55)	FFQ	SSB: ≥151.4 mL/2000 kcal/day vs. none FJ: ≥120 vs. < 9.4 mL/2000 kcal/day	RR: 1.07 (0.82,1.41) RR: 1.08 (0.83,1.41)	Age, sex, smoking, BMI, EI, time on study, race, family history of pancreatic cancer, intake of red, and processed meat.
Navarrete-Muñoz et al., 2016 [45]	10 European countries ^†^, EPIC	Pancreatic	Cohort	477,206 11.4	865	51	F (70)	DQ- country specific	SB: >196.4 vs. 0.1–13.1 mL/day SB: increase by 100 mL/day SSB: >121.4 vs. 0.1-4.5 mL/day SSB: increase by 100 mL/day ASB: >92.2 vs. 0.1-2.0 mL/day ASB: increase by 10 mL/day FJ ^6^: >123.1 vs. 0.1-8.3 mL/day FJ ^6^: increase by 100 mL/day	HR: 0.90 (0.68–1.19) HR: 1.02 (0.98–1.06) HR: 0.90 (0.65–1.25) HR: 1.02 (0.97–1.08) HR: 0.99 (0.61–1.60) HR: 1.02 (0.96–1.08) HR: 0.74 (0.57–0.97) HR: 0.91 (0.84–0.98)	Age, sex, smoking, BMI, alcohol, EI, study center, PA, and DM. FJ and SB were mutually adjusted.
Schernhammer et al., 2005 [46]	US, HPFS, NHS	Pancreatic	Cohort	136,587 14 HPFS, 20 NHS	379	53.7	F (65)	FFQ	SSB: <143.6 vs. > 11.2 mL/day ASB: <143.6 vs. > 11.2 mL/day	RR: 1.13 (0.81–1.58) RR: 1.02 (0.79–1.32)	Age, sex, smoking, BMI, follow-up cycle, PA, DM, and other soft drink intake.
**Genitourinary Cancers (Prostate, Renal Cell, Urinary Bladder, Urothelial Cell)**											
**Source**	**Country, Study Name**	**Cancer Type**	**Study Design**	**Population Follow-Up (Years)**	**Cases**	**Age (Mean/SD or Range)**	**Sex (%)**	**Dietary Assessment Method**	**Type and Amount of Beverages Intake ^+^**	**HR/RR/OR** **(95% CI)**	**Adjustments**
Bruemmer et al., 1997 [60]	US	Bladder	PB case-control	620	215	45–65	M (62)	Interview	SSB: >240 vs. < 8 mL/day	OR: 0.4 (0.2–1.1) (M) OR: 5.7 (1.2–26.9) (F) OR: 1.6 (0.7–3.6) (M) OR: 2.3 (0.8–6.3) (F)	Age, country, and smoking.
									ASB: >240 < 8 mL/day		
De Stefani et al., 2007 [61]	Uruguay	Bladder	HB case-control	756	255	30–89	M (88)	64-item FFQ	SB: ≥142 vs. <142 mL/day	OR: 1.1 (0.7–1.7)	Age, sex, residence, education, familiar history of UBC, BMI, occupation, smoking, intake of mate, coffee, tea, and milk.
Hemelt et al., 2010 [62]	China	Bladder	HB case-control	792 3	400	65.8	M (79)	DQ	SB: consumers vs. none FJ: daily vs. none	OR: 2.01 (1.10–3.68) OR: 0.66 (0.26–1.66)	Age, sex, smoking, and frequency and duration of smoking.
Radosavljević et al., 2003 [63]	Serbia	Bladder	HB case-control	260	130	64.9	M (79)	101-item FFQ	SB: >15.7 mL/day (mean) vs. none FJ: >11.6 mL/day (mean) vs. none	OR: 4.73 (2.72–8.18) OR: 0.30 (0.18–0.50)	Smoking
Turati et al., 2015 [64]	Italy	Bladder	HB case-control	1355	665	67 (median)	M (76)	DQ	SB ^2^: ≥47 mL/day vs. none	OR: 1.04 (0.73–1.49)	Age, sex, study center, year of interview, smoking, education, alcohol, BMI, and family history of UBC and cystitis.
Wang, 2013 [65]	US	Bladder	HB case-control	2306	1007	64.4	M (78)	FFQ	SB: ≥255.6 mL/day vs. none SSB: ≥126 mL/day vs. none ASB: ≥309.6 mL/day vs. none	OR: 1.34 (1.05–1.70) OR: 1.27 (1.02–1.58) OR: 1.06 (0.85–1.32)	Age, sex, ethnicity, EI, and smoking.
Chazelas et al., 2019 [23]	France	Prostate	Cohort	101,257 5.1 (median)	291	42.2/4.4	M (100)	24H-DR	SFJ: >141.7 vs. <46.1 mL/day (cut-off) SFJ: increase by 100 mL/day SSB: >65.5 vs. <14.0 mL/day (cut-off) SSB: increase by 100 mL/day ASB: >7.9 vs. <2.7 mL/day (cut-off) ASB: increase by 10 mL/day FJ: >97.8 vs. <19.9 mL/day (cut-off) FJ: increase by 100 mL/day	HR: 1.39 (0.96–2.02) HR: 1.10 (0.92–1.31) HR: 1.19 (0.83–1.72) HR: 1.24 (0.95–1.62) HR: 1.33 (1.01–1.75) HR: 0.57 (0.24–1.34) HR: 1.04 (0.76–1.42) HR: 0.97 (0.79–1.2)	Smoking, education, PA, BMI, and height.
Drake et al., 2012 [35]	Sweden, MDC	Prostate	Cohort	8128 14.9	817	45–73	M (100)	168-item FFQ, 7-d menu book Interview	SSB: 297.8 mL/day (median) vs. none FJ: 200 mL/day (median) vs. none	HR: 1.13 (0.92–1.38) HR: 0.99 (0.81–1.22)	Age, year of study entry, time of data collection, EI, height, WC, PA, smoking, education, birth in Sweden, alcohol, calcium and selenium intake, and risk by death from all causes except PC.
Ellison et al., 2000 [36]	Canada, NCSS	Prostate	Cohort	3400 23	201	50–84	M (100)	FFQ	SB ^2^: ≥100 mL/day vs. none SB ^2^: ≥any vs. none	RR: 1.29 (0.74–2.26) RR:1.09 (0.78–1.35)	Age, alcohol, smoking, BMI, fiber, and EI.
Hodge et al., 2018 [54]	Australia, MCCS	Prostate	Cohort	35,593 19	433	54.6	M (100)	121-item FFQ	SSB: ≥200 vs. <6.7 mL/day ASB: ≥200 vs. <6.7 mL/day	HR: 1.08 (0.78–1.50) HR: 0.81 (0.49–1.33)	Socioeconomic indexes, country of birth, alcohol, smoking, PA, and Med-diet score. ASB also for SSB consumption and WC.
Jain et al., 1998 [66]	Canada	Prostate	PB case-control	1253	617	69.8	M (100)	Q-DH	SB ^2^: >200 mL/day vs. none	OR: 0.79 (0.53–1.17)	Age, EI
Makarem et al., 2018 [52]	US	Prostate	Cohort	3184 4	157	54.3	M (100)	FFQ	SFJ: >401 vs. <212.1 mL/day (cut-off) SSB: >180 vs. <25.7 mL/day (cut-off) FJ: >180 vs. <48.9 mL/day (cut-off)	HR: 1.06 (1.03–1.09) HR: 1.38 (0.80–2.38) HR: 1.03 (1.01–1.06)	Age, smoking, BMI, EI, alcohol, PA, education, WC, DM, CVD, antioxidant use, and energy from fat and diet soda intake.
Miles et al., 2018 [31]	US	Prostate	Cohort	22,720 9	1996	65.6 (5.9)	M (100)	FFQ	SSB: >183 vs. <6 mL/day (cut-off) FJ: >190 vs. <24 mL/day (cut-off)	HR: 1.21 (1.06–1.39) HR: 1.07 (0.94–1.22)	Age, sex, smoking, BMI, EI, DM, education, race, family history of PC, and PSA screens.
Sharpe et al., 2002 [67]	Canada	Prostate	PB case-control	875	399	61.5	M (100)	Interviews or DQ	SB ^7^: daily drank vs. never drank weekly	OR: 1.0 (0.7–1.4)	Age, ethnicity, socioeconomic status, BMI, cumulative cigarette smoking, and alcohol.
Hodge et al., 2018 [54]	Australia, MCCS	Renal cell	Cohort	35,593 19	146	54.6	M/F	121-item FFQ	SSB: ≥200 vs. <6.7 mL/day ASB: ≥200 vs. <6.7 mL/day	HR: 1.48 (0.87–2.53) HR: 0.92 (0.46–1.84)	Socioeconomic indexes, country of birth, alcohol, smoking, PA, Med-diet score, and sex. ASB also for SSB consumption and WC
Hu et al., 2009 [68]	Canada	Renal cell	PB case-control	6177	1138	20–80	M (51)	FFQ	SB: >230 mL/day vs. none SB: increase by 230 md J: >236 vs. ≤23 mL/day J: increase by 118 mL/day	OR: 1.26 (0.96–1.67) OR: 1.05 (0.97–1.13) OR: 1.53 (1.18–1.99) OR: 1.08 (1.04–1.13)	10-year age groups, province, education, BMI, sex, EI, smoking, intake of alcohol meat, vegetables, and fruits.
Lee et al., 2006 [37]	US	Renal cell	Cohort	136,587 14 HPFS 20 NHS	248	53.7	F (65)	FFQ	SB: ≥670 vs. <47.9 mL/day SSB: increase by 335 mL/day ASB: increase by 335 mL/day FJ: increase by 335 mL/day	RR: 1.03 (0.64–1.68) RR: 0.95 (0.69–1.31) RR: 0.97 (0.82–1.15) RR: 1.06 (0.88–1.28)	BMI, EI, alcohol, smoking, history of HT, DM, multivitamin use, and parity.
Maclure and Willet, 1990 [70]	US	Renal cell	PB case-control	430	203	30–>80	M (67)	FFQ	SB: >480 vs. <68.6 mL/day ASB: >480 vs. <68.6 mL/day FJ: ≥ 480 vs. ≤ 34.3 mL/day	OR: 2.6 (1.4–4.8) OR: 2.7 (1.1–6.5) OR: 0.56 (0.22–1.4)	Age, sex, body weight/height, EI, and education
Ros et al., 2011 [38]	10 European countries ^†^, EPIC	Urothelial cell	Cohort	233,236 9.3	513	25–70	F (71)	DQ-country specific	SB: ≥99 vs. <8 mL/day (M); ≥20 vs. <8 mL/day (F) FJ: ≥72 vs. <8 mL/day (M); ≥79 vs. 8 mL/day (F)	HR: 1.03 (0.83–1.30) HR: 1.32 (1.05–1.66)	Smoking, EI from fat and nonfat sources. Stratified by age at entry, sex, and center.
	**Gynecological Cancers (Cervical, Endometrial, Epithelial Ovarian, Ovarian)**										
**Source**	**Country, Study Name**	**Cancer Type**	**Study Design**	**Population Follow-Up (Years)**	**Cases**	**Age (Mean/SD or Range)**	**Sex (%)**	**Dietary Assessment Method**	**Type and Amount of Beverages Intake ^+^**	**HR/RR/OR** **(95% CI)**	**Adjustments**
Herrero et al., 1991 [71]	Colombia, Costa Rica, Mexico and Panama	Cervical	HB, PB case-control	2033	622	46.5	F (100)	FFQ	FJ: >240 vs. <0.8 mL/day	OR: 0.90 (0.7–1.2)	Age, study site, age at 1st intercourse, number of sexual partners and pregnancies, presence of HPV 16/18, interval since last Pap smear, and number of household facilities.
Verreault et al. 1989 [72]	US	Cervical	PB case-control	416	189	20–74	F (100)	66-items FFQ	FJ: ≥ 355 vs. ≤ 48 mL/day	RR: 0.3 (0.2–0.6)	Age, education, smoking, frequency of Pap smears, use of barrier and OC, history of cervical-vaginal infection, age at first intercourse, and number of sexual partners.
Inoue-Choi et al., 2013 [39]	US	Endometrial type I	Cohort	23,039 14	506	61.6	F (100)	FFQ	SFJ: >424.3 vs. ≤55.7 mL/day SSB: >87.4 mL/day vs. none ASB: >144 mL/day vs. none FJ: >288 vs. ≤20.6 mL/day SFJ: >424.3 vs. ≤55.7 mL/day SSB: >87.4 mL/day vs. none ASB: >144 mL/day vs. none FJ: >288 vs. ≤20.6 mL/day	HR: 1.48 (1.09–2.00) HR: 1.78 (1.32–2.40) HR: 0.77 (0.59–1.01) HR: 1.16 (0.87–1.56) HR: 1.09 (0.55–2.15) HR: 1.31 (0.63–2.69) HR: 0.89 (0.48–1.68) HR: 0.97 (0.50–1.88)	Age, smoking, BMI, PA, alcohol, HRT, age at menarche and at menopause, number of live births, DM, and coffee intake.
		Endometrial type II			89						
Hodge et al., 2018 [54]	Australia, MCCS	Endometrial	Cohort	35,593 19	167	54.6	F (100)	121-item FFQ	SSB: ≥200 vs. <6.7 mL/day ASB: ≥200 vs. <6.7 mL/day SSB: ≥200 vs. <6.7 mL/day ASB: ≥200 vs. <6.7 mL/day	HR: 1.02 (0.54–1.91) HR: 0.81 (0.42–1.55) HR: 1.35 (0.71–2.56) HR: 1.37 (0.72–2.61)	Socioeconomic indexes, country of birth, alcohol, smoking, PA, Med-diet score, and sex. ASB also for SSB consumption and WC.
		Ovarian			130						
King et al., 2013 [73]	US	Epithelial ovarian	PB case-control	595 7	205	>21	F (100)	FFQ and Interview	SSB: ≥151.2 vs. <21.6 mL/2000 kcal/day SSB: increase by 360 mL/day	OR: 1.31 (0.77–2.24) OR: 1.63 (0.94–2.83)	Age, education, race, age at menarche, MS, parity, OC use, HRT, BMI, smoking, PA, DM, tubal ligation, intake of fiber, fat, and saturated fat.
Leung et al., 2016 [74]	Canada	Epithelial ovarian	PB case-control	2111 11	524	40–79	F (100)	FFQ and Interview	SB: >9.9 mL/day vs. none	OR: 0.97 (0.72–1.31)	Age, race, education, BMI, smoking, alcohol, history of ovarian/breast cancer, OC use, parity, MS, HRT, and study site.
Song et al., 2008 [75]	US	Epithelial ovarian	PB case-control	2050 3	781	35–74	F (100)	FFQ	SB ^3^ (caff): ≥720 mL/day vs. none SB ^3^ (not caff): ≥720 mL/day vs. none	OR: 1.51 (1.03–2.22) OR: 2.60 (1.25–5.39)	Age, BMI, education, smoking, race, country, years of diagnosis, number of pregnancies, OC use, hysterectomy, and family history of breast/ovarian cancer.
	**Hepatobiliary Cancers (Biliary Tract, Gallbladder, Liver)**										
**Source**	**Country, Study Name**	**Cancer Type**	**Study Design**	**Population Follow-Up (Years)**	**Cases**	**Age (Mean/SD or Range)**	**Sex (%)**	**Dietary Assessment Method**	**Type and Amount of Beverages Intake ^+^**	**HR/RR/OR** **(95% CI)**	**Adjustments**
Stepien et al., 2014 [28]	10 European countries ^†^, EPIC	Biliary tract	Cohort	477,206 11.4	236	51	F (70)	DQ-country specific	SB: 282.9 mL/day vs. none FJ ^1^: 171.7 mL/day vs. none SB: 282.9 mL/day vs. none FJ ^1^: 171.7 mL/day vs. none SB: 282.9 mL/day vs. none SB: increase by 300 mL/wk SSB: increase by 330 mL/wk ASB: increase by 330 mL/wk FJ ^1^: 171.4 mL/day vs. none FJ ^1^: increase by 200 mL/wk	HR: 0.96 (0.90–1.00) HR: 0.99 (0.95–1.03) HR: 0.97 (0.90–1.06) HR: 1.04 (1.00–1.08) HR: 1.83 (1.11–3.02) HR: 1.05 (1.02–1.07) HR: 1.00 (0.95–1.06) HR: 1.06 (1.03–1.09) HR: 1.38 (0.80–2.38) HR: 1.03 (1.01–1.06)	BMI, alcohol, EI, PA, DM, and education.
		IHBT			66						
		HCC			191						
Larsson et al., 2016 [49]	Sweden, SMC, COSM	IHBT EHBT Gallbladder	Cohort	70,832 13.4	21 127 71	45–83	M (56)	96-item FFQ	SB: ≥400 mL/day vs. none SB: ≥400 mL/day vs. none SB: ≥400 mL/day vs. none	HR: 1.69 (0.41–7.03) HR: 1.79 (1.02–3.13) HR: 2.24 (1.02–4.89)	Age, sex, education, smoking, BMI, dietary protein intake, and EI.
	**Hematologic Cancers (Leukemia, Lymphoma, Myeloma)**										
**Source**	**Country, Study Name**	**Cancer Type**	**Study Design**	**Population Follow-Up (Years)**	**Cases**	**Age (Mean/SD or Range)**	**Sex (%)**	**Dietary Assessment Method**	**Type and Amount of Beverages Intake ^+^**	**HR/RR/OR** **(95% CI)**	**Adjustments**
Schernhammer et al., 2012 [24]	US, HPFS, NHS	Leukemia	Cohort	136,587 14 HPFS 20 NHS	339	53.7	F (65)	FFQ	SSB: ≥335 mL/day vs. none ASB: ≥335 mL/day vs. none SSB: ≥335 mL/day vs. none ASB: ≥335 mL/day vs. none SSB: ≥335 mL/day vs. none ASB: ≥335 mL/day vs. none	RR: 1.06 (0.56–2.00) RR: 1.42 (1.00–2.02) RR: 1.47 (0.76–2.83) RR: 1.29 (0.89–1.89) RR: 1.34 (0.98–1.83) RR: 1.13 (0.94–1.34)	Age, BMI, EI, PA, alcohol, race, fruit and vegetables consumption, menopause, and HT. SSB were adjusted for use of ASB and vice-versa.
		Multiple myeloma			285						
		NHL			1324						
McCullough et al., 2014 [40]	US, CPS-II NCH	NHL	Cohort	100,442 10	1196	47–95	F (57)	Willett FFQ	ASB: >355 mL/day vs. none SSB: >355 mL/day vs. none	RR: 0.92 (0.73–1.17) RR: 1.10 (0.77–1.58)	Education, race, WC, PA, BMI, EI, DM, family history of cancer, HTR and NSAIDs use, cholesterol-lowering medication, intake of alcohol, read and processed meat, milk, saturated fat, fruits and vegetables, and tea and coffee.
	**Upper Aerodigestive Cancers (Larynx, Oral Cavity, Oropharyngeal Squamous Cell, Pharynx)**										
**Source**	**Country, Study Name**	**Cancer Type**	**Study Design**	**Population Follow-Up (Years)**	**Cases**	**Age (Mean/SD or Range)**	**Sex (%)**	**Dietary Assessment Method**	**Type and Amount of Beverages Intake ^+^**	**HR/RR/OR** **(95% CI)**	**Adjustments**
Zvrko et al., 2008 [82]	Montenegro	Larynx	HB case-control	216 2	108	59.9 (9.7)	M (82)	DQ	SB: yes vs. no	OR: 0.38 (0.16–0.92)	Age, sex, smoking, alcohol, coffee, diet, personal and familiar medical history, education, housing and work conditions, and exposure to toxic components.
Ren et al., 2010 [34]	US, NIH-AARP-DHS	Larynx Pharynx Oral cavity	Cohort	481,563 2	307 178 391	50–71	M (59)	124-item FFQ	SB: ≥355 vs. ≤355 mL/day SB: ≥355 vs. ≤355 mL/day SB: ≥355 vs. ≤355 mL/day	HR: 0.82 (0.55–1.23) HR: 0.76 (0.46–1.25) HR: 0.77 (0.54–1.09)	Age, sex, smoking, alcohol drinking, BMI, EI, education, ethnicity, PA, intake of fruit, vegetables, and red and white meat.
Lissowska et al., 2003 [83]	Poland	Oral cavity	HB case-control	246	122	23–80	M (64)	25-item DQ	FJ: >57 vs. <28.6 mL/day	OR: 0.35 (0.15–0.80)	Age, sex, residence, drinking, and smoking habit.
Kreimer et al., 2006 [84]	9 countries ^‡^, IARC-MOCS	OOSC	HB case-control	3402	1670	NR	M/F	FFQ	FJ: height vs. low intake	OR: 0.8 (0.6–1.1)	Age, sex, country, education, BMI, smoking, chewing, and alcohol.
	**Other Cancers**										
**Source**	**Country, Study Name**	**Cancer Type**	**Study Design**	**Population Follow-Up (Years)**	**Cases**	**Age (Mean/SD or Range)**	**Sex (%)**	**Dietary Assessment Method**	**Type and Amount of Beverages Intake ^+^**	**HR/RR/OR** **(95% CI)**	**Adjustments**
Vincenti et al., 2008 [85]	Italy	Cutaneous melanoma	PB case-control	118	59	56	F (53)	188-item FFQ	FJ (no OJ): increase by 10 mL/day OJ: increase by 10 mL/day	RR: 0.95 (0.87–1.03) RR: 0.94 (0.88–1.00)	EI, family history of melanoma, skin type, history of sunlight exposure, and sunburns.
Dubrow et al., 2012 [47]	US	Glioma	Cohort	545,771 10	904	62.8 (median)	M (60)	FFQ	SB: >720 mL/day vs. none	HR: 0.87 (0.65–1.15)	Age, sex, race, EI, height, fruit and vegetables intake, and nitrite intake from plants
Luqman et al., 2014 [86]	Pakistan	Lung	HB case-control	1200	400	<40–>70	M (73)	DQ	J: yes vs. no	OR: 0.3 (0.3–0.4)	Not reported
Wu A. et al., 1997 [87]	US	Small intestine	PB case-control	1034	36	30–65	M (69)	Interview	SSB ^7^: daily vs. never	OR: 3.6 (1.3–9.8)	Age, ethnicity, and sex.
Zamora-Ros et al., 2018 [48]	10 European countries ^†^, EPIC	Thyroid	Cohort	477,206 11.4	748	51	F (70)	DQ- country specific	FJ ^1^: > 94 vs. < 1 mL/day FJ ^1^: increase by 50 mL/day	HR: 1.23 (0.98–1.53) HR: 1.02 (0.99–1.06)	Age, sex, smoking status, BMI, EI, alcohol, PA, education, center, menopausal status and type, OC use, and infertility problems.
**Overall Cancers**											
**Source**	**Country, Study Name**	**Cancer Type**	**Study Design**	**Population Follow-Up (Years)**	**Cases**	**Age (Mean/SD or Range)**	**Sex (%)**	**Dietary Assessment Method**	**Type and Amount of Beverages Intake ^+^**	**HR/RR/OR** **(95% CI)**	**Adjustments**
Bassett et al., 2020 [50]	Australia, MCCS	Non-obesity related *	Cohort	35,109 19	4789	27–76	F (61)	121-item FFQ	SSB: >375 vs. none or < 12.5 mL/day ASB: >375 vs. none or < 12.5 mL/day	HR: 1.02 (0.86–1.21) HR: 1.23 (1.02–1.48)	Alcohol, country of birth, Med-diet score, PA, socio-economic position, sex, and smoking. ASB also adjusted for SSB intake.
Makarem et al., 2018 [52]	US	Breast, Colorectal, Prostate	Cohort	3184 4	565	54.3	F (53)	FFQ	SFJ: >501 vs. <73.2 mL/day SSB:>180 mL/day vs. none FJ: >216 vs. <23 mL/day (cut-off)	HR: 1.28 (0.97–1.70) HR: 1.00 (0.79–1.27) HR: 1.05 (0.80–1.38)	Age, sex, EI, alcohol, smoking, and BMI.
Hodge et al., 2018 [54]	Australia, MCCS	Obesity-related	Cohort	35,593 19	3283	54.6	F (100)	121-item FFQ	SSB: ≥200 vs. <6.7 mL/day ASB: ≥200 vs. <6.7 mL/day	HR: 1.14 (0.93–1.39) HR: 1.00 (0.79–1.27)	Socioeconomic indexes, country of birth, alcohol, smoking, PA, Med-diet score, and sex. ASB also for SSB consumption and WC.
Chazelas et al., 2019 [23]	France, NNS	Breast, Colorectal, Prostate	Cohort	101,257 5.1 (median)	2193	42.2/14.4	F (78)	24H-DR	SFJ: >141.7 vs. <46.1 mL/day (cut-off) SFJ: increase by 100 mL/day SSB: >65.5 vs. <14.0 mL/day (cut-off) SSB: increase by 100 mL/day ASB: >7.9 vs. <2.7 mL/day (cut-off) ASB: increase by 10 mL/day FJ: >97.8 vs. <19.9 mL/day (cut-off) FJ: increase by 100 mL/day	HR: 1.30 (1.17–1.52) HR: 1.18 (1.10–1.27) HR: 1.06 (1.02–1.21) HR: 1.19 (1.08–1.32) HR: 1.00 (0.84–1.19) HR: 1.02 (0.94–1.10) HR: 1.14 (1.01–1.29) HR: 1.12 (1.03–1.23)	Smoking, education, PA, BMI, and height.

^+^ Expressed in milliliter (mL) per day (d) or week (wk) or none (nonconsumers). ^†^ Denmark, France, Germany, Greece, Italy, Norway, Spain, Sweden, The Netherlands, and the United Kingdom. ^‡^ Italy, Spain, Poland, Northern Ireland, India, Cuba, Canada, Australia, and Sudan. * All identified cancers except esophagus (adenocarcinoma), pancreas, colorectum, breast (post-menopausal), endometrium, kidney, ovary, gallbladder, liver, gastric cardia, meningioma, thyroid, multiple myeloma. ^1^: Fruit juice and vegetables juice. Vegetables juice <2%. ^2^: Colas. ^3^: Colas and root beer. ^4^: Not carbonated beverages. ^5^: Sugarcane juice (20.3%), honeydew melon juice (14.1%), apple juice (12.8%), watermelon juice (9%), carrot juice (9%), pineapple juice (6.4%), star fruit juice (5.1%), and lemon juice drink (5.1%). The remaining canned grape, tomato, and prune juice, along with papaya, plum, and fresh celery juice, each comprised 1.3–2.6% of the total juice consumption reported. ^6^: Fruit juice and nectars. ^7^: Carbonated beverages. AA: African American; AEGJ: adenocarcinoma of the esophagus-gastric junction; ASB: artificially sweetened beverages; BC: breast cancer; BF: breastfeeding; BMI: body mass index; BWHS: Black Women’s Health Study; Caff: caffeinated; CI: confidence interval; COSM: Cohort of Swedish Men; CPS-NCS: Cancer and Prevention Study, Nutrition Cohort Study; CR: colorectal; CVD: cardiovascular disease; EA: European American; EAC: esophageal adenocarcinoma; EI: energy intake; EPIC: European Prospective Investigation into Cancer and nutrition; DH: diet history; DM: diabetes mellitus; DQ: dietary questionnaire; 24H-DR: 24 h dietary recall; F: female; FFQ: food frequency questionnaire; FJ: natural fruit juice; GI: glycemic index; HB: hospital-based; HCC: Hepatocellular Carcinoma; HCS: Hokkaido Cohort Study; HPFS: Health Professionals Follow-up Study; HPV: Human Papilloma Virus; HR: hazard ratio; HRT: hormone replacement therapy; HT: hypertension; IARC-MOCS: International Agency for Research on Cancer, Multicenter Oral Cancer Study; IHBT: intrahepatic biliary tract; J: natural fruit and vegetable juice; M: male; MCCS: Melbourne Collaborative Cohort Study; MDC: Malmö Diet and Cancer; Med: Mediterranean; MS: menopausal status; NCFD: not carbonated fruit drinks; NCSC: Nutrition Canada Survey Study; NHL: non-Hodgkin lymphoma; NNS: Nutri Net-Santé; NIH-AARP-DHS: National Institute of Health-American Association of Retired Persons, Diet and Health Study; NSAIDs: nonsteroidal anti-inflammatory drugs; NHS: Nurses’ Health Study; OC: oral contraceptive; OJ: orange juice; OOSC: oral and oropharyngeal squamous cell; OR: odds ratio; PA: physical activity; PB: population-based; PC: prostate cancer; Post-M: post-menopausal breast cancer; PSA: prostate-specific antigen; Pre-M: pre-menopausal breast cancer; Q: quantitative; Q1: first quartile; Q4: quartile four; RR: relative risk; SB: total sweetened beverages, sugar and artificially sweetened beverages; SCC: squamous cell carcinoma SCHS: the Singapore Chinese Health Study; SD: standard deviation; SFQ: structured food questionnaire; SFB: San Francisco Bay Study: SFJ: beverages high in sugar, added or natural, SSB + FJ; SSB: sugar-sweetened beverages; SMC: Swedish Mammography Cohort; SQ: semiqualitative; SUVIMAX: Supplementation en Vitamines et Mineraux Antioxydants Study; UBC: urinary bladder cancer; UP: ultraprocessed; US: the United States; WC: waist circumference; WCHS: Women’s Circle of Health Study.

**Table 2 nutrients-13-00516-t002:** Summary of the results of the meta-analysis (random effects model).

Cancer Type	Exposure	N° of Studies		RR (95% CI)	*I^2^* (%)	Tau^2^	*p* within Group ^+^	95% PI
		Cohort	Case-Control					
Breast	SSB	4	3	1.14 (1.01−1.30)	0.0	0.0073	0.69	0.88, 1.47
Breast	FJ	3	0	1.13 (0.93−1.38)	0.0	0.0017	0.79	0.52, 2.46
Breast Pre-M	SSB	3	2	1.37 (0.99−1.88)	55.7	0.0358	0.06	0.68, 2.76
Breast Post-M	SSB	4	2	1.18 (0.79−1.75)	54.8	0.1080	0.05	0.43, 3.23
Colorectal	SSB	4	0	1.18 (0.99−1.41)	0.0	0.0039	0.71	0.82, 1.69
Colorectal	FJ	2	2	0.79 (0.16−3.87)	88.5	0.8629	<0.001	0.008, 73.94
Colorectal *	FJ	2	1	1.29 (0.78−2.12)	0.0	0.0120	0.63	0.17, 9.81
Colorectal	SB	0	3	2.02 (0.45−9.01)	62.9	0.2711	0.07	0.00, 5753.1
Colorectal *	SB	0	2	1.57 (0.74−3.35)	0.0	0.0010	0.67	–
Bladder	SB	0	5	1.66 (0.78−3.56)	83.4	0.3226	<0.001	0.22, 12.37
Bladder *	SB	0	4	1.27 (0.85−1.90)	25.3	0.0425	0.26	0.45, 3.60
Prostate	SSB	5	0	1.18 (1.10−1.27)	0.0	0.0012	0.92	1.03, 1.35
Prostate	FJ	4	0	1.03 (1.01−1.05)	0.0	0.0001	0.93	0.98, 1.09
Prostate	SB	1	2	0.97 (0.56−1.69)	2.9	0.0241	0.36	0.07, 12.7
Renal cell	SB	1	2	1.44 (0.46−4.50)	65.4	0.1559	0.056	0.00, 604.16
Pancreatic	SB	4	4	1.28 (0.95−1.72)	58.6	0.0962	0.02	0.56, 2.90
Pancreatic	SSB	4	2	1.01 (0.92−1.11)	0.0	0.0016	0.92	0.87, 1.17
Pancreatic	ASB	3	2	1.07 (0.77−1.48)	43.6	0.0480	0.13	0.48, 2.36

* Results excluding outliers; ^+^
*p* values of Cochran’s Q-test heterogeneity. ASB: artificial sweetened beverage; FJ: fruit juice; PI: prediction intervals; Post-M: post-menopausal; Pre-M: pre-menopausal; RR: risk ratio; CI: confidence interval; SB: sweetened beverage (including both SSBs and ASBs); SSB: sugar-sweetened beverage.

## Data Availability

Data for doing the meta-analysis is available in the manuscript’s tables.

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
