# Peer review of "Consumption of Sweet Beverages and Cancer Risk. A Systematic Review and Meta-Analysis of Observational Studies"

_nutrients, 2021, doi:10.3390/nu13020516_

Round 1

Reviewer 1 Report

Consumption of Sweet Beverages and Cancer Risk. A Systematic Review and Meta-Analysis of Observational Studies

This manuscript presents a large body of published data and careful consideration of the results. The systematic review part is very good. The meta-analytical part of the manuscript does not answer the main questions of all meta-analyses:

  1. Taking into account that the estimates of the association between an exposure and a disease vary between the studies, can the estimates be summarized?
  2. The estimates can be summarized if the evidence of heterogeneity between the studies is low.
  3. If heterogeneity between the studies is high, it is important to reveal the sources of such heterogeneity. Usually, this is done with meta-regression. For example, we can find that the estimates are greater in populations with a high proportion of other risk factors for cancer or where the westernized diet is more prevalent. This would be a very valuable insight.

The usual problems in meta-analyses that are found in this study are the following:

  1. Wrong interpretation of I2 as an index of heterogeneity.
  2. When heterogeneity is found (even with the correct indices), the random effect summary is presented only. Instead, the studies should be stratified by some characteristics that can explain heterogeneity, and stratified summary estimates should be reported with insights from meta-regression.

The meta-analytical part of this manuscript should be significantly revised accordingly.   

Reviewer 2 Report

The role of sweetened beverages in damaging health is very relevant and deserves attention. It appears however that evidence concerning cancer risk often remains controversial.

I was surprised by the observation that fruit juices are associated with an increase in cancer risk: is it due to the addition of sugar also in these products?

I think it is very important to state (as the authors do) that sweetened beverages increase the risk of cardiometabolic diseases: very often we see reviews on cancer risk that completely forget other (more severe) diseases.

I would like the Authors to make a comment about the clinical relevance of their observations beyond the strict limits of statistics. In many articles in recent years only statistical evaluation is considered relevant, but I think other aspects may also be of interest.

The suggestion of a “homogeneous classification of beverages” is very important!

page 1 line 34: I wonder if reference 1 is suitable to support the increase in consumption beverages.

             line 42: please explain what you mean by “empty calories”

page 2   lines 54-56 are repeated in lines 64-65

lines 65-67: I think also including the role of colorants will distract from the main topic of the review.

I think the table is very informative and represents a point of strength of this review.

page 22 line 435 “including the reduce of SSB intake” better “including a reduced SSB intake”

References to be amended

7

14

103: there is no indication on how to access this document.

Round 2

Reviewer 1 Report

I think this is publishable as it is.